# Healing from Within: How Gut Microbiota Predicts IBD Treatment Success—A Systematic Review

**DOI:** 10.3390/ijms25158451

**Published:** 2024-08-02

**Authors:** Luana Alexandrescu, Alina Doina Nicoara, Doina Ecaterina Tofolean, Alexandra Herlo, Andreea Nelson Twakor, Cristina Tocia, Anamaria Trandafir, Andrei Dumitru, Eugen Dumitru, Cristian Florentin Aftenie, Ionela Preotesoiu, Elena Dina, Ioan Tiberiu Tofolean

**Affiliations:** 1Gastroenterology Department, “Sf. Apostol Andrei” Emergency County Hospital, 145 Tomis Blvd., 900591 Constanta, Romania; alexandrescu_l@yahoo.com (L.A.); cristina.tocia@yahoo.com (C.T.); dr.andreidumitru@gmail.com (A.D.); eugen.dumitru@yahoo.com (E.D.); elena.dina@ymail.com (E.D.); tofoleanioan@yahoo.com (I.T.T.); 2Medicine Faculty, “Ovidius” University of Constanta, 1 Universitatii Street, 900470 Constanta, Romania; tofoleandoina@yahoo.com (D.E.T.); animaria.trandafir@yahoo.com (A.T.); afteniecristian@gmail.com (C.F.A.); ionelia.phb@yahoo.com (I.P.); 3Internal Medicine Department, “Sf. Apostol Andrei” Emergency County Hospital, 145 Tomis Blvd., 900591 Constanta, Romania; andreea.purcaru@365.univ-ovidius.ro; 4Pneumology Department, “Sf. Apostol Andrei” Emergency County Hospital, 145 Tomis Blvd., 900591 Constanta, Romania; 5Department XIII, Discipline of Infectious Diseases, “Victor Babes” University of Medicine and Pharmacy Timisoara, 2 Eftimie Murgu Square, 300041 Timisoara, Romania; alexandra.mocanu@umft.ro; 6Academy of Romanian Scientist, 3 Ilfov Street, 050044 Bucharest, Romania

**Keywords:** Crohn’s disease, ulcerative colitis, dysbiosis, fecal microbiota, inflammatory bowel diseases, inflammation, biological therapies, FMT

## Abstract

Recent research indicates that the microbiome has a significant impact on the progression of inflammatory bowel disease (IBD) and that creating therapies that change its composition could positively impact the outcomes of IBD treatment. This review summarizes the results of extensive studies that examined IBD patients undergoing several therapies, including anti-TNF medication, vedolizumab, ustekinumab, probiotics, and fecal microbiota transplantation (FMT), and the alterations in their gut microbiota’s composition and function. The objective was to investigate the variety and effectiveness of microbial species in order to discover new biomarkers or therapeutic targets that could improve the outcome of treatment for these patients. This research aimed to offer useful insights into personalized medicine techniques for managing IBD. Beneficial bacteria such as *Faecalibacterium prausnitzii* and *Roseburia* have been consistently linked to favorable clinical outcomes, whereas pathogenic bacteria such as *Escherichia coli* and *Clostridioides difficile* are associated with worsening disease conditions. Although many studies have examined the role of gut microbiota in IBD, there is still a need for more targeted research on the connection between specific microbial communities and treatment outcomes. This study sought to address this gap by exploring the intricate relationship between the gut microbiota composition and the effectiveness of IBD medications.

## 1. Introduction

The intestine is the home for a diverse range of bacteria, viruses, and fungus, together known as the intestinal microbiota, which thrive in this natural environment [1]. This microbiota is dependent on the host’s gut for survival, has certain metabolic processes, and aids in several physiological functions [2]. Furthermore, the gut microbiota significantly influences the host’s metabolism, development, and immune system [3].

The two main IBDs, Crohn’s disease (CD) and ulcerative colitis (UC), are long-lasting digestive conditions characterized by immune system-related inflammation, the cause of which is still unclear [4,5]. Symptoms include diarrhea, abdominal pain, and blood in the stool [6,7]. Recent studies suggest that this condition may be linked to an imbalance in intestinal microorganisms or immunomediated factors [8,9,10,11].

Technological advancements, such as metagenomics and meta transcriptomics, have allowed for the identification and validation of the genetic basis of the physiological activities of the gut microbiota [12,13]. Despite this significant progress, the cause of IBD remains unknown [14]. The increased prevalence in identical twins as opposed to fraternal twins, and within specific families, indicates a genetic factor in the development of the disease [15,16]. Furthermore, a group of genome-wide association studies, formally known as GWAS, have discovered a multitude of genetic variants, providing additional evidence for a genetic component of IBD [17,18]. Nevertheless, genetics alone is seldom sufficient to initiate the disease [19]. The occurrence of IBD has been increasing in parallel with the progress of the modern world, lifestyle changes, and the evolution of society [20]. This phenomenon has occurred within a brief timeframe, and its occurrence cannot be solely attributed to genetic alterations or the process of natural selection [21]. When examining the causes of these disorders, it is important to examine other variables such as environmental aspects, shifts in dietary patterns from plant-based to animal-based processed meals, the increased prevalence in smoking from a younger age, and antibiotic use [22].

Sir Samuel Wilks, a pioneering physician in the late 19th century, originally proposed the concept that transmissible bacterial pathogens could be the underlying cause of IBD [23]. He was also the first to use the phrase “ulcerative colitis” to describe a condition that closely resembles the current understanding of UC [23].

Research has demonstrated that patients with active IBD experience substantial alterations in the composition of their microbiome, such as an increase in the dominant phylotype *Proteobacteria* and a decrease in Firmicutes [24,25,26,27]. These discoveries have caused a fundamental change in the approach to treating this condition, focusing on the goal of restoring the equilibrium of microorganisms in the gut [28,29]. Specifically, in IBD, there is a lower variety of microorganisms and an increased presence of dangerous bacteria [30,31,32]. Moreover, the literature indicates that altering the gut microbiota through therapies such as prebiotics/probiotics and FMT can have a beneficial effect on the outcomes of IBD [33]. This personalized approach not only improves the effectiveness of medications but also reduces side effects, providing a more accurate strategy, focused on the patient, for controlling IBD [34]. 

Existing reviews often focus on single treatments or limited patient cohorts, failing to provide a holistic view of microbial dynamics across different therapies and populations. Additionally, there is a lack of detailed comparative analyses that incorporate recent advancements in microbiome research and methodologies. This review aimed to bridge this gap by systematically evaluating the impact of diverse treatments on gut microbiota composition, considering both beneficial and harmful bacterial shifts, across a wide range of IBD studies.

## 2. Materials and Methods

### 2.1. Literature Search

A thorough examination of the medical literature published from January 2018 to June 2024 was conducted using the databases PubMed and ScienceDirect. The objective was to document the microbiome composition in Crohn’s disease and ulcerative colitis as well as factors that might contribute to the effectiveness of medical treatment, with a focus on biological therapies.

#### Search Strategy

Boolean operators were used, and they included subject headings and keywords such as “*ulcerative colitis*”, “*Crohn’s disease*”, “*inflammatory bowel disease*”, “*IBD biological therapies*”, “*IBD mechanism of action*”, *and “gut microbiome*.” The term “AND” was used to combine different groups of search terms, ensuring comprehensive results.

### 2.2. Inclusion and Exclusion Criteria

#### 2.2.1. Inclusion Criteria

Patients of any age diagnosed with inflammatory bowel disease.

Studies that included microbiome analysis (using any method) either before or after the treatment.

Studies related to advanced therapy for IBD.

Studies that provided clear definitions of therapeutic response.

#### 2.2.2. Exclusion Criteria

Non-peer-reviewed research.

Case studies.

Studies with insufficient data or without quantifiable results for outcomes.

Abstracts, papers not accessible in English.

Research examining non-validated medical treatments or ileal pouch anal anastomose

Animal studies.

### 2.3. Study Selection and Rationale for Choices

The study selection process consisted of two stages:

Stage 1: Screening of Titles and Abstracts: Initial screening was performed to identify relevant studies based on titles and abstracts.

Stage 2: Full-Text Assessment and Data Extraction: A thorough assessment of the full-text articles was conducted. 

#### Rationale for Methodological Choices 

Databases: PubMed and ScienceDirect were chosen for their comprehensive coverage and the high-quality medical literature.

PICO Framework: This framework was chosen to systematically structure the review process, allowing for a clear comparison of different interventions and outcomes [35].

PRISMA Guidelines: Employed to ensure transparency in reporting the systematic review [36].

### 2.4. Resolving Discrepancies in Data Extraction and Study Selection

During Stage 1 (screening titles and abstracts), discrepancies between reviewers were resolved through a consensus approach. Initially, two reviewers independently screened the titles and abstracts of all identified studies. Any discrepancies at this stage were discussed between the two reviewers to reach an agreement. If a consensus could not be reached, a third reviewer was consulted to provide an additional perspective. For Stage 2 (full-text assessment and data extraction), two reviewers independently evaluated each article and extracted relevant data. Any inconsistencies or disagreements in data extraction were again resolved through discussion and consensus between the two reviewers. If necessary, the senior author (Dr. Luana Alexandrescu) was involved to make the final decision (please see Appendix A).

### 2.5. Systematic Review Framework

Population (P): Patients of any age diagnosed with IBD.

Intervention (I): Methods of analysis of microbiota.

Comparison (C): Gut microbiota of IBD patients before and after the intervention or comparison with healthy controls.

Outcome (O): Establish the relationship between microbiome and IBD and the connection between microbiome and the success of medical treatment.

### 2.6. Focus of the Study

The study focused on the following advanced therapies for IBD: thiopurines, methotrexate, anti-TNF therapy, anti-integrin therapy, ustekinumab, risankizumab, and JAK inhibitors. Additionally, studies investigating the use of antibiotics, 5-aminosalicylic acid (5ASA) treatment, or corticosteroids were reviewed.

### 2.7. Prisma Framework

A total of 1680 citations were retrieved after scanning the aforementioned databases. After eliminating duplicate entries and excluding 82 items that did not satisfy the search parameters, the list was reduced to 375 remaining articles. 

Based on the abstracts, 224 studies were excluded from this research, as they did not meet the criteria. Additionally, 82 papers were eliminated because they did not have the necessary data for extraction and analysis. Furthermore, 20 studies were excluded due to the follow-up period being too short to assess the long-term outcomes. Another 18 studies were omitted because they were commentary or editorial rather than original research. Lastly, 14 articles were disregarded, as the full text was not available. Thus, the final analysis was based on a total of 17 search results that met the criteria for this investigation (Table 1). 

Filters applied for PubMed search: Free full text, meta-analysis, randomized controlled trial, systematic review, English, from 1 January 2018–30 June 2024.

The search resulted in a total of 47 citations for “gut microbiota and IBD” available on PubMed and 381 available on ScienceDirect. 

The search for “IBD and biological therapies” led to 101 articles on PubMed and 435 articles on ScienceDirect.

The search for “gut microbiome and biological therapies” revealed a total of 195 citations on PubMed and 521 on ScienceDirect.

Following the process of screening and conducting a full-text review, a total of 17 papers were selected for the qualitative analysis (please see Appendix A).

Figure 1 presents the PRISMA flow diagram, which is essential for detailing the selection process of the studies included in this review. This diagram visually represents the identification, screening, eligibility, and inclusion phases of this systematic review process. The relevance of Figure 1 lies in its ability to provide a transparent methodology explaining how the final set of studies was determined.

### 2.8. Statistical Analysis

Key studies from this systematic research include those of Vich Vila et al. (2018) [37], which assessed gut microbiota composition in IBD patients treated with anti-TNF therapy, and Ananthakrishnan et al. (2017) [38], which explored gut microbiota as a predictor of response to vedolizumab in IBD patients. Franzosa et al. (2019) [39] studied the effects of ustekinumab on gut microbiota, while Sokol et al. (2020) [40] evaluated FMT to maintain remission in CD patients. Huang et al. (2023) [41] reviewed the use of probiotics for treating UC, and Costello et al. (2019) [42] evaluated FMT as a treatment strategy for this disease. Ribaldone et al. (2019) [43] examined microbiome changes within 6 months of adalimumab therapy in CD patients, while He et al. (2021) [44] correlated clinical aspects with microbiome composition in UC patients.

Further studies included those of Crothers et al. (2021) [45], which evaluated daily oral FMT for maintaining remission in UC, and Olaisen et al. (2021) [46], which investigated the bacterial mucosa-associated microbiome in CD patients. Lloyd-Price et al. (2019) [47] conducted a multi-omics analysis of the gut microbial ecosystem in IBD, while Coufal et al. (2019) [48] examined inflammation and gut barrier markers in these patients. Pittayanon et al. (2020) [49] and Forbes et al. (2018) [50] performed a meta-analysis on gut microbiota differences in IBD patients and healthy individuals. Nikolaus et al. (2017) [51] investigated tryptophan metabolism in IBD patients, and Fornelos et al. (2020) [52] studied the effects of N-acylethanolamines on their gut bacteria. Lastly, Rausch et al. (2023) [53] analyzed fecal microbial communities before anti-inflammatory treatments in CD and UC patients. 

Figure 2, below, illustrates the sensitivity analysis conducted to determine the effect of removing each individual study on the pooled effect size.

The blue dots represent the pooled effect size estimates when each study was excluded from the analysis, while the red lines show the corresponding 95% confidence intervals. The sensitivity analysis demonstrated the strength of the overall findings, as the pooled effect size estimates remained relatively stable across the removal of different studies. Notably, the exclusion of Sokol et al. [40] resulted in a slightly higher pooled effect size (red colour), indicating its influence on the overall estimate. On the other hand, the removal of Crothers et al. [45] (green colour) resulted in a lower pooled effect size. However, the overall consistency, regardless of which study was removed, suggested that the results were not excessively dependent on any single study. 

Figure 3 presents a heatmap of the risk of bias analysis across the studies, assessing six categories: selection bias, performance bias, detection bias, attrition bias, reporting bias, and other biases. It identifies areas of potential bias, helping to critically assess the reliability of the findings. The color gradient ranges from blue (low risk) to red (high risk), providing a visual representation of the bias levels. Most studies, such as those by Vich Vila et al. [37] and Ananthakrishnan et al. [38], display a low risk of bias across all categories. However, studies such as those by Huang et al. [41] and Pittayanon et al. [49] show higher risks in specific categories, such as selection and performance bias, suggesting areas of potential methodological weakness. The uniform blue seen in many cells points out the general reliability of the findings, while the red and gray areas highlight where caution should be applied when interpreting results. 

The subplots included in Figure 4 provide a comprehensive overview of the demographic characteristics of the study populations. Thus, Figure 4 includes bar charts, as follows: the top left bar chart reveals a wide range of participant numbers, with Huang et al. [41] using the largest sample size (1120 participants), while Crothers et al. [45] used the smallest sample size (12 participants). This disparity in sample sizes highlights the different scopes and scales of the studies, ranging from extensive reviews to focused pilot studies. 

The bar chart on the top right illustrates the gender distribution within these studies. The percentage of male participants varied, with the study by Crothers et al. [45] having the highest male percentage, at 67%, followed closely by the study by Ribaldone et al. [43], at 60%. Most other studies demonstrated a more balanced gender distribution, reflecting efforts to ensure demographic representation. The addition of the study by Rausch et al. (2023) [53] showed a male percentage of 59%, which aligns with the trend of maintaining gender balance across studies.

The bottom left bar chart shows the female participant percentages, highlighting that most studies maintained a near-equal gender balance, further emphasizing inclusivity. Studies such as those by Fornelos et al. [52], Nikolaus et al. [51], and Forbes et al. [50] achieved a perfect 50/50 gender split, emphasizing the commitment to demographic diversity.

The bottom right scatter plot with error bars describes the age characteristics of participants, showing the mean age and standard deviation for each study. Notably, Ribaldone et al. [43] reported the highest mean age of participants, at 52.5 years, suggesting a focus on an older cohort, whereas Sokol et al. [40] and He et al. [44] targeted slightly younger populations. The inclusion of Rausch et al. [53], with a mean participant age of 42.07 years and a standard deviation of 17.14 for their study, indicated a broad age range within this study, reflecting its diverse participant pool.

## 3. Results 

These studies presented in Table 1 offer distinct perspectives on the correlation between microbiota and its impact on the efficacy of biological therapy. Table 2 consolidates data to highlight consistent patterns, such as the decrease in beneficial bacteria like *Faecalibacterium prausnitzii*, *Roseburia*, *Bacteroides*, *Ruminococcaceae*, and *Lachnospiraceae* in IBD patients, which are important for their anti-inflammatory properties and role in maintaining gut health (Vich Vila et al. [37], Ananthakrishnan et al. [38], Franzosa et al. [39], Olaisen et al. [46], Pittayanon et al. [49]).

The table also allows for an easy comparison of findings across studies, emphasizing the increase in harmful bacteria like *Escherichia coli*, *Clostridioides difficile*, *Enterobacteriaceae*, and *Proteobacteria*. According to studies conducted by Vich Vila et al. [37], Franzosa et al. [39], and Coufal et al. [48], this shift towards dysbiosis exacerbates inflammation and disease severity. The presence of these bacteria correlates with increased disease activity and highlights their role in driving intestinal inflammation.

Equally, there is a marked decrease in beneficial bacteria, such as Faecalibacterium prausnitzii, Roseburia, Bacteroides, Ruminococcaceae, Lachnospiraceae, Firmicutes, and Bacteroidetes, in IBD patients (Vich Vila et al. [37], Ananthakrishnan et al. [38], Franzosa et al. [39], Olaisen et al. [46], Pittayanon et al. [49]). 

The table also groups studies examining the impact of FMT and other therapeutic interventions on gut microbiota. FMT responders showed an increase in beneficial bacteria such as *Faecalibacterium prausnitzii*, *Roseburia* spp., and *Ruminococcus* spp., which are linked to positive clinical outcomes and remission maintenance in IBD patients (Sokol et al. [40], Costello et al. [42], Crothers et al. [45]). Notably, Ribaldone et al. [43] highlighted that *Faecalibacterium prausnitzii*, *Ruminococcus gnavus*, *Escherichia coli*, and *Bacteroides ovatus* decreased in FMT responders, emphasizing the complex dynamics of gut microbiota in response to therapy.

Table 3 details the specific therapies used in various studies, the key bacteria that were increased or decreased, and additional notes on study populations and methodologies. This table is essential as it outlines the microbiota changes linked to specific treatments like anti-TNF therapy, vedolizumab, ustekinumab, and FMT, providing a nuanced understanding of how different therapies impact gut microbiota.

Significant increases were noted in *Faecalibacterium prausnitzii*, *Roseburia*, and other members of the *Ruminococcaceae* family across several studies [37,40,43] especially following treatments such as anti-TNF therapy, FMT, and probiotics. For instance, Sokol et al. [40] and Costello et al. [42] found that FMT led to an increase in *Faecalibacterium prausnitzii* and *Roseburia*, which are linked to positive clinical outcomes and remission maintenance in IBD patients.

Probiotic treatments have also led to increased levels in *Lactobacillus* and *Bifidobacterium* species. Huang et al. [41] demonstrated that various probiotic strains could modulate gut microbiota, reduce inflammation, and improve intestinal barrier function, further supporting their therapeutic potential in managing IBD. On the other hand, pathogenic bacteria, including *Escherichia coli*, *Clostridioides difficile*, and other *Proteobacteria*, are regularly decreased after successful treatments [40,44]. 

Ribaldone et al. [43] also observed a decrease in *Proteobacteria* in CD patients treated with adalimumab, indicating a positive shift towards a healthier gut microbiota composition. Similarly, Rausch et al. [53] reported that successful FMT led to a decrease in harmful bacteria such as *Escherichia coli* and *Clostridioides difficile*, further emphasizing the therapeutic potential of microbiota modulation in IBD management.

The data across multiple studies reinforce the importance of gut microbiota as both a marker and mediator of health in IBD, suggesting that future treatments should continue to focus on microbiota modulation to achieve better patient outcomes [37,40,41,42,44,53].

Figure 5 summarizes the impact of various therapies on gut microbiota composition. It visually demonstrates the contrasting changes in important bacteria, providing a quick reference for understanding which therapies promote beneficial bacteria and which reduce harmful bacteria.

As it can be seen in Figure 5, FMT and various treatments (including mesalamine, corticosteroids, and immunosuppressants) resulted in a greater increase in beneficial bacteria, indicating their positive impact on gut microbiota composition. 

Table 4 shows the comparison of beneficial and harmful bacteria in IBD patients highlights significant differences in the microbial composition associated with disease progression. Notably, beneficial bacteria such as *Faecalibacterium prausnitzii*, *Roseburia*, and *Bifidobacterium* are frequently linked with anti-inflammatory effects and the production of SCFAs, which are crucial for maintaining gut health. Studies such as those by Vich Vila et al. [37] and Sokol et al. [40] emphasize the increase in these beneficial bacteria in response to treatments such as anti-TNF therapy and FMT, indicating their role in promoting remission in IBD patients.

Conversely, the table also identifies several harmful bacteria associated with IBD, such as *Bacteroides*, *Ruminococcus gnavus*, and *Escherichia coli*. Their presence is frequently associated with an imbalance in microbiome inflammation, as indicated in research conducted by Ananthakrishnan et al. [38] and Franzosa et al. [39]. For example, *Ruminococcus gnavus* is linked to inflammatory complexes that can exacerbate IBD. Furthermore, the excessive presence of *Escherichia coli* and *Clostridioides difficile* is linked to intestinal infections and exacerbated illness outcomes. Moreover, *Lachnospiraceae* and *Ruminococcaceae* can be both helpful or harmful, depending on the circumstances. The dual roles of these bacteria are highlighted in recent studies by Ribaldone et al. [43] and Rausch et al. [53]. 

Ribaldone et al. [43] demonstrated that adalimumab therapy in CD patients led to a significant increase in Firmicutes and Bacteroidetes and a decrease in Actinobacteria and Proteobacteria. This shift in microbial composition is associated with improved clinical outcomes. Similarly, Rausch et al. [53] found that anti-inflammatory treatments in IBD patients increased the abundance of beneficial bacteria such as *Faecalibacterium prausnitzii*, *Roseburia*, and *Bifidobacterium*, while harmful bacteria such as *Escherichia coli* and *Clostridioides difficile* decreased, further supporting the literature [70,71] on the importance of microbiota modulation in managing IBD.

### Importance of Analyzing the Microbiome in IBD

According to “World Gastroenterology Organization Global Guidelines” on probiotics and prebiotics from February 2023 [72], studies on the use of probiotics in CD have shown no significant benefits for inducing or maintaining remission of the condition. These findings suggest that probiotics are not effective in the long-term management of the disease. In line with these conclusions, Sokol et al. [40] focused on FMT rather than probiotics, leading to the conclusion that FMT helps maintain remission and positively alters gut microbiota composition in CD patients.

Despite the popularity of probiotics for various digestive issues, the current evidence does not support their use as an effective treatment for inducing or maintaining remission in Crohn’s disease [73].

On the other hand, the same report highlighted some promising findings regarding the use of probiotics in the treatment of UC [72]. Individual studies included in “World Gastroenterology Organization Global Guidelines” indicated that certain probiotics might be safe and potentially as effective as conventional therapies in achieving response and remission rates for patients with mild to moderately active UC; this finding was applicable to both adults and pediatric populations. These findings suggest a potential role for probiotics as an alternative treatment option in specific cases of ulcerative colitis. This study also reflects these promising findings. Thus, the experimental research by Huang et al. [41] on the use of probiotics for treating UC led to the conclusion that various probiotic strains modulate gut microbiota, reduce inflammation, and improve intestinal barrier function.

However, it is important to consider the broader context of these findings. According to a meta-analysis conducted by Kaur et al. [74], the efficacy of probiotics for inducing remission in mild to moderate UC is not sufficient. Many studies reviewed in this analysis, such as those by Vich Vila et al. [37] and Ananthakrishnan et al. [38], show positive correlations between gut microbiota composition and treatment outcomes but also note the need for more rigorous, long-term studies to validate these findings. 

## 4. Discussions

The intricate link between gut microbiota and IBD has received significant attention, especially in terms of understanding the distinct responses to various treatments. 

Vich Vila et al. [37] investigated the composition of the microbiome in IBD patients undergoing anti-TNF therapy. Their investigation revealed an increase in good bacteria, such as *Faecalibacterium prausnitzii* and *Roseburia*, among responders. These findings align with previous studies. For example, Franzosa et al. [39] similarly observed elevated levels of *Roseburia* and *Ruminococcaceae* after administering ustekinumab. These taxonomic groups have been extensively studied for their ability to reduce inflammation and produce butyrate, crucial for maintaining a healthy gut [75,76]. 

Ananthakrishnan et al. [38] highlighted the role of the gut microbiome in predicting responses to vedolizumab, noting increased *Streptococcus salivarium* in responders. This aligns with Costello et al. [42], who observed an increase in beneficial bacteria such as *Anaerofilum pentosovorans* and *Ruminococcaceae* post-FMT, suggesting that successful therapy is often marked by a rise in beneficial butyrate-producing bacteria. These bacteria help maintain the integrity of the intestinal barrier and possess anti-inflammatory properties [77]. Mayorga et al. [78] found a correlation between gut microbiota diversity and IBD severity, with reduced diversity linked to more severe disease. Sokol et al. [40] demonstrated that FMT significantly altered the gut microbiota composition in CD patients, increasing the prevalence of *Roseburia* and *Faecalibacterium prausnitzii*. These results align with findings by Huang et al. [41], who reviewed the effects of various probiotics in UC, highlighting strains such as *Lactobacillus reuteri* and *Bifidobacterium longum* for their beneficial impact on gut microbiota and inflammation reduction. Li et al. [79] also found that probiotics significantly improve clinical symptoms in IBD patients, further supporting their beneficial role.

Conversely, several studies identified harmful bacteria associated with IBD activity. Vich Vila et al. [37] noted higher levels of *Bacteroides* and *Ruminococcus gnavus* in non-responders to anti-TNF therapy. He et al. [44] found increased *Proteobacteria* and *Escherichia-Shigella* in UC patients, linked to heightened inflammation and gut barrier disruption. These bacteria’s pro-inflammatory roles are well documented, with *Proteobacteria* often acting as pathobionts that exacerbate IBD symptoms [80]. Gilliland et al. [81] also noted that the presence of these bacteria is linked to higher illness severity in patients with IBD.

Nikolaus et al. [51] highlighted the association between increased tryptophan metabolism and IBD activity, finding higher levels of *Escherichia coli* and *Clostridioides difficile* in active disease phases. In line with these findings, Khorsand et al. [82] revealed that individuals with active CD and UC had raised levels of *Enterobacteriaceae* and *Fusobacteriaceae*, providing more evidence of their detrimental effects.

Fornelos et al. [52] examined the effects of N-acylethanolamines on gut bacteria, discovering that these substances have varying effects on bacterial proliferation in individuals with IBD. They noted a rise in pathogenic bacteria, specifically *Escherichia coli* and *Bacteroides vulgatus*, suggesting that changes in gut metabolic conditions can impact disease development. Coufal et al. [48] corroborated these findings, observing comparable microbial alterations in IBD. Nieva et al. [83] demonstrated that targeted dietary interventions to manipulate gut microbiota can effectively regulate disease activity in patients with IBD.

A study conducted by Pittayanon et al. [49] showed an increase in bad bacteria, specifically *Proteobacteria* and *Fusobacteria*. This agrees with other studies in the literature, as Sorboni et al. [84] highlighted the significance of microbiota regulation in the management of IBD, identifying the presence of the same harmful bacteria. Furthermore, Vujkovic-Cvijin et al. [85] observed a similar dysbiosis as a characteristic feature of IBD.

The comparative analysis of these studies highlights the dual role of specific bacterial taxa in IBD, with beneficial bacteria often promoting anti-inflammatory effects and maintaining gut health while harmful bacteria exacerbate inflammation and disrupt gut barrier function. This duality is crucial for understanding therapeutic outcomes and developing targeted interventions [86]. Restoring a balanced gut microbiota through therapies such as probiotics, FMT, and microbial-targeted treatments holds promise for managing IBD more effectively.

Future research should prioritize large-scale, long-term studies with standardized methodologies to validate these findings and elucidate the mechanisms through which gut microbiota modulate IBD. 

### Limitations

Many of the studies reviewed had relatively small sample sizes and limited diversity in the participant populations. For instance, Vich Vila et al. [37] included 150 participants with diverse ethnic backgrounds, but the majority had moderate disease, which may not fully represent the broader IBD patient population. Similarly, other studies, such as those by Ribaldone et al. [53] and Crothers et al. [45], included only 12 participants, limiting the generalizability of the findings. 

The reviewed studies employed different methodologies, including observational studies, randomized controlled trials, and systematic reviews. This variability can introduce heterogeneity in the results, making direct comparisons challenging. For example, Sokol et al. [40] conducted a randomized controlled study on FMT, while Huang et al. [41] reviewed experimental research on probiotics, which can lead to differences in outcome measures and interpretations.

The studies reviewed used various therapeutic interventions such as anti-TNF therapy, vedolizumab, ustekinumab, probiotics, and FMT. These interventions have different mechanisms of action and may affect the gut microbiota in distinct ways, complicating the comparison of their effects on bacterial composition. For example, Ananthakrishnan et al. [38] focused on vedolizumab, whereas Costello et al. [42] investigated the effects of FMT, highlighting the need for caution when generalizing findings across different treatments.

Several studies focused on particular bacterial groups, possibly neglecting other crucial components of the gut microbiota that could have major impacts on IBD. For example, while studies frequently highlighted *Faecalibacterium prausnitzii* and *Roseburia* as beneficial, less attention was devoted to other beneficial bacteria that might also contribute to disease modulation. This selective focus can lead to an incomplete understanding of the microbiome’s overall impact on IBD.

Several studies had relatively short follow-up periods, limiting the ability to assess the long-term effects of the interventions on gut microbiota and clinical outcomes. For instance, Costello et al. [42] evaluated the effects of FMT on 8-week remission, but longer-term data are needed to determine the sustainability of these changes.

There is a potential for publication bias, as studies with positive results are more likely to be published than those with negative or inconclusive findings. This bias can skew the overall understanding of the role of gut microbiota in IBD and the effectiveness of microbiome-targeted therapies.

## 5. Conclusions

This comprehensive study features the significant role of gut microbiota in the onset and management of IBD. Numerous studies have pinpointed key bacteria, including *Faecalibacterium prausnitzii* and *Roseburia*, which are vital for maintaining gut health and improving IBD symptoms.

Conversely, detrimental bacteria, including *Escherichia coli* and *Clostridioides difficile*, were often linked to exacerbated clinical results. Therapies such as anti-TNF, vedolizumab, and fecal microbiota transplantation have shown promise in favorably altering the gut microbiota composition, enhancing treatment efficacy and patient prognosis.

Future research should prioritize large-scale, long-term studies with standardized methodologies to validate these findings and elucidate the mechanisms through which gut microbiota modulate IBD. Addressing these research gaps will be crucial for advancing microbiome-based therapeutic strategies and improving patient outcomes in IBD. Ultimately, the composition and predictability of gut microbiota have an intricate and multidimensional role in the effectiveness of treating IBD. Future studies should also focus on longitudinal analyses using advanced sequencing techniques to uncover potential biomarkers. In conclusion, a more profound comprehension of the interaction between gut microbiota and treatments for IBD has the potential to lead to groundbreaking therapeutic approaches that specifically focus on the microbiome.

## Figures and Tables

**Figure 1 ijms-25-08451-f001:**
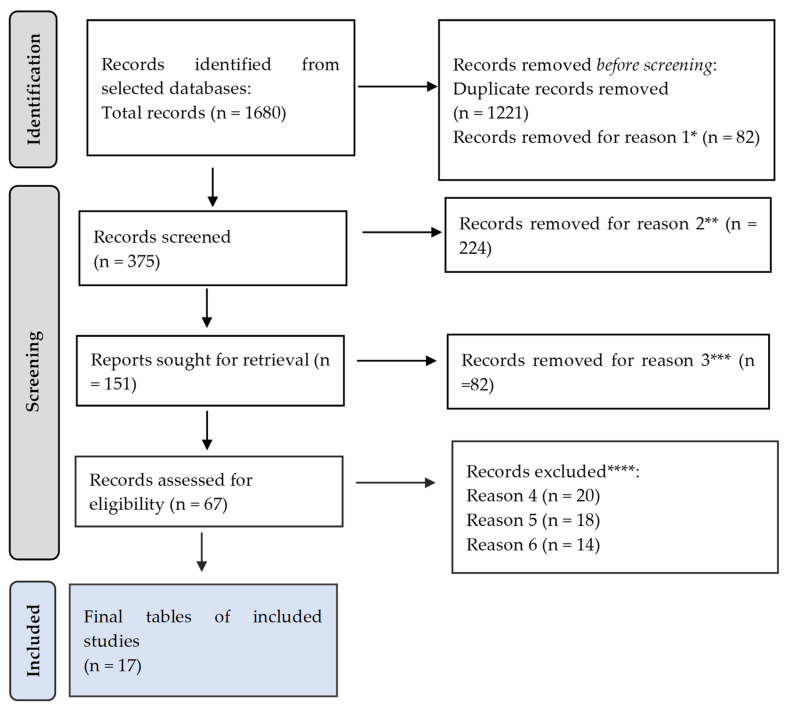
PRISMA framework. Reason 1 *: The study does not utilize the appropriate design for the review. Reason 2 **: The study does not meet the minimum quality threshold based on the assessment criteria. Reason 3 ***: The study lacks necessary data for extraction and analysis. Reason 4 ****: The study’s follow-up period was too short to assess the long-term outcomes. Reason 5 ****: The study is a review, commentary, or editorial rather than original research. Reason 6 ****: Full text is not available.

**Figure 2 ijms-25-08451-f002:**
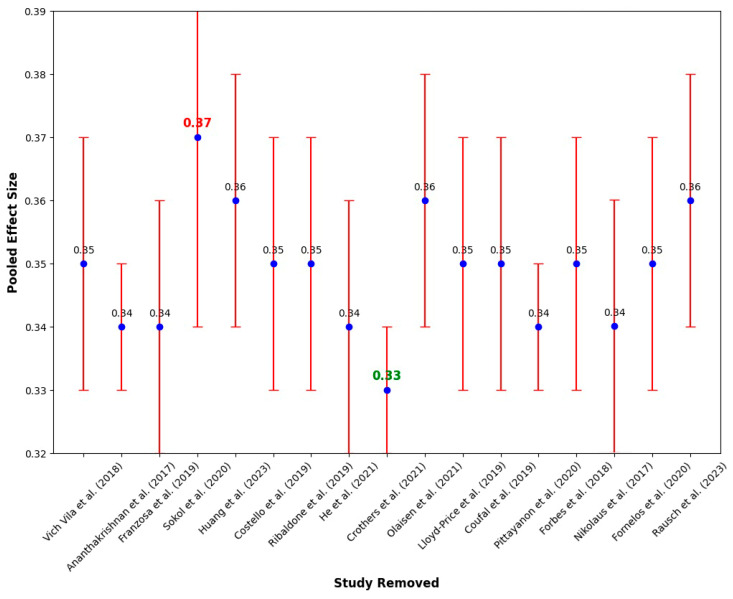
Sensitivity analysis of the 17 studies [37,38,39,40,41,42,43,44,45,46,47,48,49,50,51,52,53].

**Figure 3 ijms-25-08451-f003:**
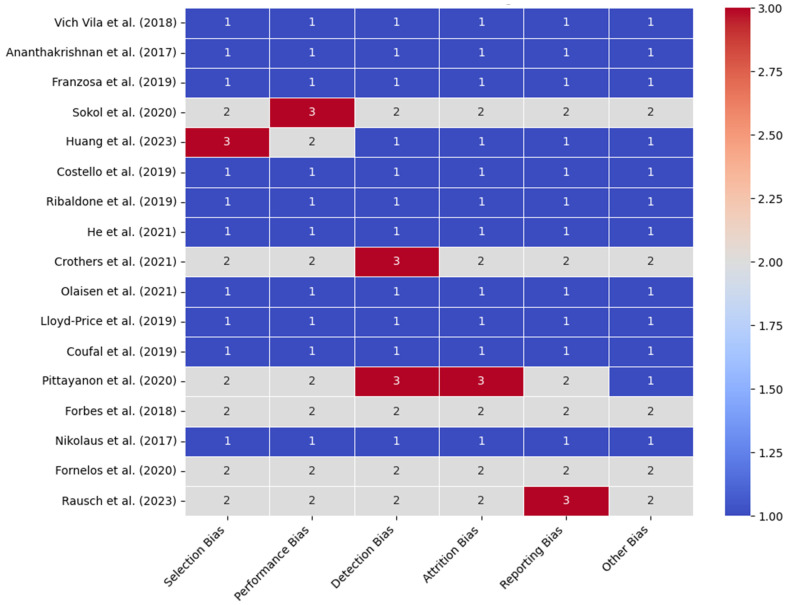
Risk bias analysis of the 17 studies [37,38,39,40,41,42,43,44,45,46,47,48,49,50,51,52,53].

**Figure 4 ijms-25-08451-f004:**
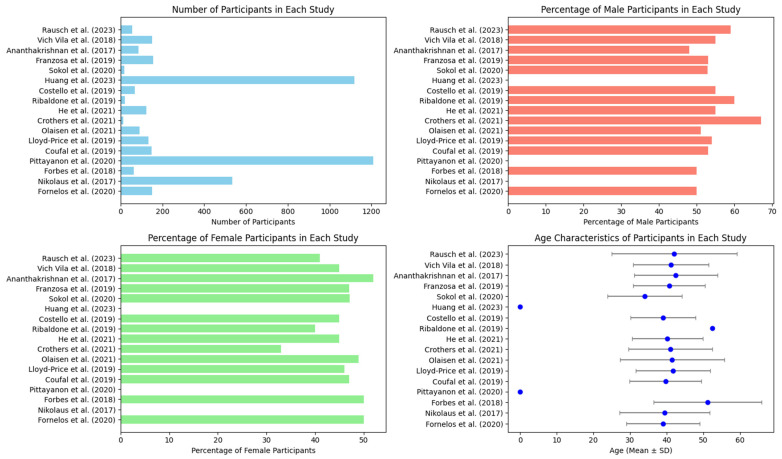
Four subplots showing the number of participants in each study, the percentage of male participants in each study, the percentage of female participants in each study, and the age characteristics (mean age ± SD) of participants in each study [37,38,39,40,41,42,43,44,45,46,47,48,49,50,51,52,53].

**Figure 5 ijms-25-08451-f005:**
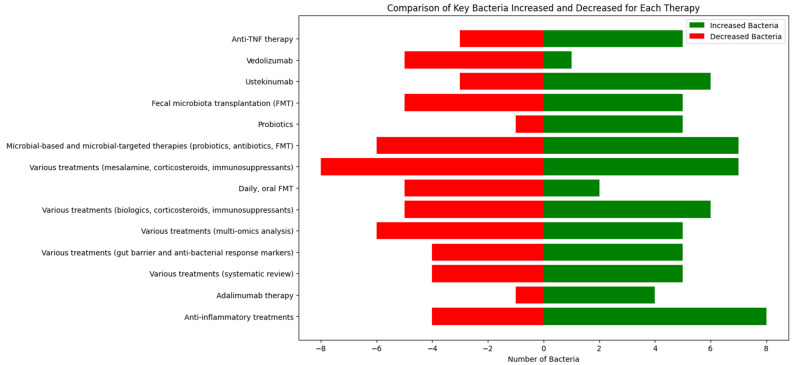
The number of key bacteria that increased (green) and decreased (red) for each therapy.

**Table 1 ijms-25-08451-t001:** Overview of study characteristics and main conclusions in gut microbiota and IBD treatment research.

Study	PICO Framework				Short Description	Main Conclusions	Participants	% Male	% Female	Age (Mean ± SD)	Other Characteristics
	Population (P)	Intervention (I)	Comparison (C)	Outcome (O)							
Vich Vila et al. (2018) [37]	IBD patients	Anti-TNF therapy	Responders vs. non-responders	Gut microbiota composition	Assessed gut microbiota composition in IBD patients treated with anti-TNF therapy	Higher microbial diversity and increased *Faecalibacterium prausnitzii* in responders	150	55%	45%	41.2 ± 10.3	Diverse ethnic backgrounds, majority with moderate disease
Ananthakrishnan et al. (2017) [38]	IBD patients	Vedolizumab	Responders vs. non-responders	Gut microbiota composition	Explored gut microbiota as a predictor of response to vedolizumab in IBD patients	Higher baseline levels of *Clostridiales* linked to better clinical responses	85	48%	52%	42.5 ± 11.4	Included both treatment-I and experienced patients
Franzosa et al. (2019) [39]	IBD and healthy controls	Ustekinumab	Pre- and post-intervention	Gut microbiota diversity	Studied effects of ustekinumab on gut microbiota in IBD patients	Restoration of gut microbiota diversity and increase in beneficial bacteria such as *Ruminococcaceae*	155	53%	47%	40.7 ± 9.8	Included some with previous biologic exposure
Sokol et al. (2020) [40]	Crohn’s disease patients	Fecal microbiota transplantation	Placebo	Remission maintenance	Evaluated FMT to maintain remission in CD patients	FMT helps maintain remission and positively alters gut microbiota composition in CD patients	17	52.9%	47.1%	34.0 ± 10.2	Included patients in clinical remission and those with a history of biologic therapy
Huang et al. (2023) [41]	Ulcerative colitis patients	Probiotics	Pre- and post-intervention	Gut microbiota composition	Reviewed experimental research on the use of probiotics for treating ulcerative colitis between 2018 and 2022	Various probiotic strains modulate gut microbiota, reduce inflammation, and improve intestinal barrier function	1120	N/A	N/A	N/A	Review of experimental research; patients were combined from 31 studies on humans
Costello et al. (2019) [42]	Ulcerative colitis patients	Fecal microbiota transplantation (FMT) or placebo via colonoscopy	Pre- and post-intervention	Gut ecology and clinical outcomes	Evaluated FMT as a treatment strategy for ulcerative colitis	FMT restores gut ecology and improves clinical outcomes	69	55%	45%	39 ± 8.9	Included severe cases, frequent hospitalizations
Ribaldone et al. (2019) [43]	CD patients	Fecal samples were collected before starting adalimumab therapy	Pre- and post-intervention	To evaluate any changes in the microbiome within 6 months of therapy with adalimumab	The study explored the modification of microbiota during adalimumab therapy in patients with CD	Firmicutes rose from 45.5 ± 5.1% to 48.9 ± 3.0%, *Bacteroidetes* from 33.5 ± 4.7% to 37.1 ± 4.0%; *Proteobacteria* fell from 15.7% to 10.3 ± 3.4%, and *Actinobacteria* increased from 2.6% to 3.0%	20	60%	40%	52.5	When adalimumab therapy was started, 90% of patients were also administered mesalazine, 60% of patients received systemic corticosteroids, and 20% took azathioprine
He et al. (2021) [44]	UC patients and healthy controls	Various treatments	Pre- and post-intervention	Microbiome composition	Correlating the clinical aspects with the composition of microbiome of UC patients	Specific microbial signatures correlated with disease severity and treatment response	122	55%	45%	40.2 ± 9.7	Predominantly Chinese cohort at different stages of the disease
Crothers et al. (2021) [45]	UC patients	Oral FMT administered daily	Placebo	To observe the long-term effects on maintaining remission	Daily evaluation with oral FMT in UC patients	Oral FMT administered every day helps in maintaining the remission of UC	12	67%	33%	41 ± 11.4	Randomized controlled trial (RCT)
Olaisen et al. (2021) [46]	CD patients and healthy controls	Various treatments	Inflamed vs. non-inflamed sites	To identify the composition of bacterial mucosa-associated microbiome in CD patients	Investigated the microbiome composition in ileum of CD patients	Differences in microbiome composition between inflamed and non-inflamed sites	91	51%	49%	41.5 ± 14.2	CD patients at various stages of the disease
Lloyd-Price et al. (2019) [47]	IBD patients	Various treatments	Multi-omics analysis	Gut microbial abundance	Multi-omics analysis of gut microbial ecosystem in IBD	Identified specific microbial and metabolic signatures associated with IBD	132	54%	46%	41.7 ± 10.2	Included genetic, metagenomic, and metabolomic data
Coufal et al. (2019) [48]	IBD patients and healthy controls	Various treatments	Inflammation and gut barrier markers	Antibacterial response to treatment	Investigated differences in inflammation, gut barrier, and specific antibacterial responses in IBD	Differences in markers of inflammation, gut barrier, and specific antibacterial responses in IBD types	147	53%	47%	39.7 ± 9.8	Included analysis of gut barrier and inflammatory markers
Pittayanon et al. (2020) [49]	IBD patients	Various treatments	IBD patients vs. healthy controls	Gut microbiota composition	Meta-analysis of studies on gut microbiota differences in IBD patients and healthy individuals	Discovered significant differences in microbiome among IBD patients with various stages of the disease	1210	N/A	N/A	N/A	Review of 48 randomized controlled trials with IBD patients
Forbes et al. (2018) [50]	IBD patients and healthy controls	None	IBD patients vs. healthy controls	Gut microbiota composition	Compared the microbiome of patients with immune-mediated inflammatory diseases with that of healthy controls	Discovered shared dysbiosis patterns in IBD and other immune-mediated inflammatory diseases	62	50%	50%	51.2 ± 14.7	Only patients with UC and CD, as well as the healthy controls, were selected. Patients with rheumatoid arthritis and multiple sclerosis were excluded.
Nikolaus et al. (2017) [51]	IBD patients	Evaluate serum levels of tryptophan and its metabolites	Healthy controls and varying IBD activity	Tryptophan metabolism	Investigated tryptophan metabolism in IBD patients compared to controls	Increased tryptophan metabolism is linked to higher disease activity in IBD	535	N/A	N/A	39.5 ± 12.3	Included patients at different stages of CD and UC
Fornelos et al. (2020) [52]	IBD patients and healthy controls	N-acylethanolamine (NAE) treatment	Gut bacteria from IBD patients vs. healthy controls	Bacterial growth effects and abundances	Studied effects of NAEs on gut bacteria and their altered abundances in IBD	NAEs differentially affect bacterial growth in IBD, reflecting altered gut microbiota	150	50%	50%	39.0 ± 10.0	Included patients with CD and UC and healthy controls
Rausch et al. (2023) [53]	CD and UC patients and healthy controls	Fecal microbial communities were assessed via 16S rRNA gene sequencing before administration of anti-inflammatory treatments	Gut bacteria from CD and UC patients vs. healthy controls	To determine whether FMT can induce remission in CD and UC patients	The study analyzed if FMT can increase remission rates compared to the control group	Significant differences between the microbiome of healthy individuals and IBD patients were found, and small differences or no differences were found between newly diagnosed, treatment-naïve UC and CD patients	56	59%	41%	42.07 ± 17.14	Many participants had undergone conventional treatments for CD and UC, such as corticosteroids and immunosuppressants, prior to enrolling in the trial. This inclusion criterion allowed us to evaluate FMT’s effectiveness in patients with different treatment histories.

**Table 2 ijms-25-08451-t002:** Abundance of bacteria in IBD patients as compared to healthy controls.

Gut Microbiota	Abundance Compared with Healthy People	Study
*Faecalibacterium prausnitzii*, *Roseburia.*, *Bacteroides*, *Ruminococcaceae*, *Lachnospiraceae.*, Firmicutes, Bacteroidetes	Decreased in IBD	Vich Vila et al. (2018) [37], Ananthakrishnan et al. (2017) [38], Franzosa et al. (2019) [39], Olaisen et al. (2021) [46], Pittayanon et al. (2020) [49]
*Escherichia coli*, *Clostridioides difficile*, *Enterobacteriaceae*, *Proteobacteria*	Increased in IBD, associated with dysbiosis and inflammation	Vich Vila et al. (2018) [37], Franzosa et al. (2019) [39], Coufal et al. (2019) [48]
Various probiotic strains, *Akkermansia*, *Ruminococcus*, *Bifidobacterium Clostridiales*, *Bacterioidetes*, *Tryptophan metabolism*, *N-acylethanolamines*	Increased in IBD, generally beneficial, modulate gut microbiota, reduce inflammation; some strains beneficial, others harmful	Huang et al. (2023) [41], He et al. (2021) [44], Lloyd-Price et al. (2019) [47], Forbes et al. (2018) [50], Nikolaus et al. (2017) [51], Fornelos et al. (2020) [52]
*Faecalibacterium prausnitzii*, *Roseburia*, *Ruminococcus*	Increased in FMT responders	Sokol et al. (2020) [40], Costello et al. (2019) [42], Crothers et al. (2021) [45]
*Faecalibacterium prausnitzii*, *Ruminococcus gnavus*, *Escherichia coli*, *Bacteroides ovatus*	Decreased in FMT responders	Ribaldone et al. (2019) [43]
*Faecalibacterium prausnitzii*, *Bacteroides fragilis*, *Roseburia*, *Eubacterium rectale*, *Clostridioides leptum*, *Lachnospiraceae*, *Bifidobacterium*, *Akkermansia muciniphila*	Decreased in IBD	Rausch et al. (2023) [53]
*Escherichia coli*, *Enterococcus faecalis*, *Clostridioides difficile*, *Streptococcus parasanguinis*	Increased in IBD, associated with dysbiosis and inflammation	Rausch et al. (2023) [53]

**Table 3 ijms-25-08451-t003:** Comparison table for therapies and microbiota changes in CD and UC.

Study	Disease	Therapy	Key Bacteria (Increased)	Key Bacteria (Decreased)	Other Notes
Vich Vila et al. [37]	IBD patients	Anti-TNF therapy	*Faecalibacterium prausnitzii*, *Roseburia*, *Bacteroides uniformis*, *Eubacterium rectale*, *Ruminococcus bromii*	*Bacteroides*, *Ruminococcus gnavus*, *Clostridioides clostridioforme*	Diverse ethnic backgrounds, majority with moderate disease
Ananthakrishnan et al. [39]	IBD patients	Vedolizumab	*Streptococcus salivarium*	*Bifidobacterium longum*, *Eggerthella*, *Ruminococcus gnavus*, *Roseburia inulinivorans*, *Veillonella parvula*	The relative abundance of all these taxa shifted in patients who achieved remission
Franzosa et al. [39]	IBD patients	Ustekinumab	*Roseburia*, *Bifidobacterium breve*, *Clostridioides symbiosum*, *Ruminococcus gnavus*, *Escherichia coli*, *Clostridioides clostridioforme*	*Roseburia hominis*, *Dorea formicigenerans*, and *Ruminococcus obeum*	Included some with previous biologic exposure
Sokol et al. [40]	Crohn’s disease patients	Fecal microbiota transplantation	*Roseburia*, *Ruminococcaceae*, *Faecalibacterium prausnitzii*, *Bifidobacterium*, *Akkermansia*	*Bacteroides*, *Escherichia coli*, *Enterococcus*, *Clostridioides difficile*, *Lachnospiraceae*	Included patients in clinical remission and those with a history of biologic therapy
Huang et al. [41]	Ulcerative colitis patients	Probiotics	*Lactobacillus reuteri*, *Lactobacillus rhamnosus*, *Bifidobacterium longum*, *Enterococcus faecium*, *Streptococcus thermophilus*	*Bacteroides*	Review of experimental research; no direct participant data
Costello et al. [42]	Ulcerative colitis patients	Fecal microbiota transplantation	*Anaerofilum pentosovorans*, *Bacteroides coprophilus*, *Methanobrevibacter smithii*, *Ruminococcaceae*, *Prevotellaceae*, *Coriobacteriaceae*	*Lachnospiraceae*, *Coriobacteriaceae*	Included severe cases, frequent hospitalizations
Ribaldone et al. [43]	CD patients	Adalimumab therapy	Firmicutes, Bacteroidetes, Actinobacteria	*Proteobacteria*	When adalimumab therapy was started, 90% of patients were also provided mesalazine, 60% systemic corticosteroids, and 20% azathioprine
He et al. [44]	Ulcerative colitis patients	Various treatments (mesalamine, corticosteroids, immunosuppressants)	*Proteobacteria*, *Lachnospira*, *Escherichia-Shigella*, *Enterococcus*, *Peptoclostridium*, *Haemophilus*, *Klebsiella*	*Alistipes*, *Bacteroides*, *Dialister*, *Escherichia-Shigella*, *Alistipes*, *Subdoligranulum*, *Roseburia*, *Ruminococcus*	Predominantly Chinese cohort; included various disease severities
Crothers et al. [45]	Ulcerative colitis patients	Daily, oral FMT	*Bifidobacteriales*, *Lactobacillales*	*Burkholderiales*, *Bifidobacteriales*, *Selenomonadales*, *Enterobacteriales*, *Lactobacillales Clostridiales*, *Bacterioidetes*	Single-center study; prospective, randomized pilot study
Olaisen et al. [46]	Crohn’s disease patients	Various treatments (biologics, corticosteroids, immunosuppressants)	*Lachnospiraceae*, *Clostridiales*, *Enterobacteriaceae*, *Escherichia-Shigella*, *Lachnospiraceae*, *Peptostreptococcaceae*	*Proteobacteria*, *Ruminococcaceae*, *Faecalibacterium*, *Bacterioidetes*, *Rhodospirillales*	Varied disease duration and severity
Lloyd-Price et al. [47]	IBD patients	Various treatments (multi-omics analysis)	*Lachnospiraceae*, *Faecalibacterium prausnitzii*, *Roseburia*, *Ruminococcus*, *Bacteroides fragilis*	*Escherichia coli*, *Bacteroides uniformis*, *Faecalibacterium prausnitzii*, *Eubacterium rectale*, *Bacteroides vulgatus*, *Roseburia intestinalis*, *Prevotella copri*	Included genetic, metagenomic, and metabolomic data
Coufal et al. [48]	IBD patients	Various treatments (gut barrier and antibacterial response markers)	*Bifidobacterium*, *Lactobacillus*, *Faecalibacterium prausnitzii*, *Roseburia*, *Ruminococcus*	*Proteobacteria*, *Escherichia coli*, *Enterobacteriaceae*, *Clostridioides difficile*	Included analysis of gut barrier and inflammatory markers
Pittayanon et al. [49]	IBD patients	Various treatments	Firmicutes, Bacteroidetes, *Verrucomicrobia*, *Actinobacteria*, *Spirochaetes*	*Fusobacteria*, *Proteobacteria*, *Actinobacteria*, *Spirochaetes*	Systematic review of gut microbiota differences in patients with vs. without IBD
Forbes et al. [50]	IBD patients	None	*Actinomyces*, *Eggerthella*, *Clostridioides III*, *Faecalicoccus*, *Streptococcus*, *Blautia*, *Intestinibacter*, *Bifidobacterium*	*Gemmiger*, *Lachnospira*, *Sporobacter*, *Asaccharobacter*, *Clostridioides IV*, *Coprococcus*, *Ruminococcus*, *Oscillibacter*	CD and UC patients only
Rausch et al. [53]	CD and UC patients	Fecal microbial communities assessed via 16S rRNA gene sequencing before anti-inflammatory treatments	*Faecalibacterium prausnitzii*, *Bacteroides fragilis*, *Roseburia*, *Eubacterium rectale*, *Clostridioides leptum*, *Lachnospiraceae*, *Bifidobacterium*, *Akkermansia muciniphila*	*Escherichia coli*, *Enterococcus faecalis*, *Clostridioides difficile*, *Streptococcus parasanguinis*	Many participants had undergone conventional treatments for CD and UC, such as corticosteroids and immunosuppressants, prior to enrolling in the trial

**Table 4 ijms-25-08451-t004:** Classification of bacteria as either beneficial or harmful with a brief explanation of their roles.

Beneficial Bacteria	Role	Harmful Bacteria	Role
*Faecalibacterium prausnitzii*, *Roseburia*	Anti-inflammatory effects, produce SCFAs [54]	*Bacteroides*	Associated with dysbiosis and inflammation in IBD [55,56]
*Bacteroides uniformis*	Play a role in maintaining gut barrier function, reducing inflammation [57]	*Ruminococcus gnavus*	Produce inflammatory compounds [58]
*Eubacterium rectale*	Produce SCFAs, maintain gut health [59,60]	*Clostridioides clostridioforme*	Pathogenic potential, associated with gut infections [61]
*Ruminococcus bromii*	Degradation of resistant starch, produce SCFAs [54]	*Bifidobacterium longum*, *Eggerthella*, *Roseburia inulinivorans*, *Veillonella parvula*	Reduced in remission; associated with gut dysbiosis and inflammation [57]
*Streptococcus salivarium*	Associated with oral and gut health, less pathogenic [62,63]	*Roseburia hominis*, *Dorea formicigenerans*, *Ruminococcus obeum*	Generally beneficial, but their decrease can indicate dysbiosis [57]
*Roseburia*, *Bifidobacterium breve*	SCFA production, immune regulation [54]	*Escherichia coli*, *Enterococcus*, *Clostridioides difficile*, *Lachnospiraceae*	Pathogenic potential, associated with gut infections and inflammation [61]
*Clostridioides symbiosum*	Beneficial in small amounts for gut health [58]	*Lachnospiraceae*, *Coriobacteriaceae*	Can be pathogenic in certain contexts, contributing to dysbiosis [57]
*Ruminococcaceae*	SCFA production, maintain gut barrier [54]	*E. coli*, *K. pneumoniae*, *Pasteurellaceae*, *Haemophilus*, *Neisseriaceae*, *Fusobacteriaceae*, *Bacteroidetes*, *E. faecalis*	Pathogenic potential, associated with gut infections [60]
*Bifidobacterium*	Support gut health, reduce inflammation [64]	*Proteobacteria*, *Escherichia-Shigella*, *Enterococcus*, *Peptoclostridium*, *Haemophilus*, *Klebsiella*	Associated with gut inflammation and dysbiosis [65]
*Akkermansia*	Maintain gut barrier, reduce inflammation [66]	*Burkholderiales*, *Selenomonadales*, *Enterobacteriales*, *Lactobacillales*, *Clostridiales*, *Bacterioidetes*	Pathogenic potential, associated with gut dysbiosis [59]
*Lactobacillus reuteri*, *Lactobacillus rhamnosus*, *Bifidobacterium longum*, *Enterococcus faecium*, *Streptococcus thermophilus*	Probiotic strains, reduce gut inflammation, improve gut barrier function [63,65]	*Proteobacteria*, *Ruminococcaceae*, *Bacterioidetes*, *Rhodospirillales*	Associated with gut inflammation and dysbiosis [58,61]
*Anaerofilum pentosovorans*, *Bacteroides coprophilus*, *Methanobrevibacter smithii*, *Prevotellaceae*, *Coriobacteriaceae*	Methanogens and other bacteria involved in maintaining gut health and reducing inflammation [60]	*Escherichia coli*, *Bacteroides uniformis*, *Faecalibacterium prausnitzii*, *Eubacterium rectale*, *Bacteroides vulgatus*, *Roseburia intestinalis*, *Prevotella copri*	Pathogenic potential in certain contexts, associated with gut dysbiosis [58,59,60,61,62]
*Clostridioides*, *E. rectale*, *Ruminococcaceae*, *Lachnospiraceae*, *Roseburia hominis*, *Erysipelotrichaceae*	SCFA production, maintain gut health [54]	*Gemmiger*, *Lachnospira*, *Sporobacter*, *Asaccharobacter*, *Clostridioides IV*, *Coprococcus*, *Ruminococcus*, *Oscillibacter*	Associated with gut dysbiosis and inflammation [65,66]
*Alistipes*, *Dialister*, *Subdoligranulum*, *Roseburia*	SCFA production, maintain gut health [54]	*Fusobacteria*, *Actinobacteria*, *Spirochaetes*	Associated with gut inflammation and dysbiosis [67,68,69]
*Actinomyces*, *Clostridioides III*, *Faecalicoccus*, *Streptococcus*, *Blautia*, *Intestinibacter*	SCFA production, maintain gut health [54]

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
