# Peer review of "Healing from Within: How Gut Microbiota Predicts IBD Treatment Success—A Systematic Review"

_ijms, 2024, doi:10.3390/ijms25158451_

Round 1

Reviewer 1 Report

Comments and Suggestions for Authors

Dear Authors,  

This review applied a systematic process, using evidence drawn from studies examining the alterations in gut microbiota composition in IBD patients who have had several therapies. The manuscript overall would strongly benefit from a careful revision of the clarity of study methodology and results. The English language should be revised and improved throughout the manuscript. A clear description of the evidence gap that this review is filling is required. No clear rationale on why this review is of importance.  

The methods are vague and not clearly described. The reviewed papers lacked a rigorous method. Need to be more explicit about how each criterion was applied, especially regarding the selection of articles. Provide more details on the search strategies, specifying the Boolean operators and any additional filters used. Clarify the process of data extraction and how discrepancies were resolved. Statistical analysis should be described in sufficient details. There is need to provide more statistical details on use of heterogeneity, sensitivity analysis, bias, and meta-regression. It would be benefit to include the number of records through selected databases in Figure 1.  

Tables should be well organized. Data extraction requires a lot of planning. The discussion is of poor quality as it does not explain the research findings and presents several possible explanations for the effects.

Comments on the Quality of English Language

English very difficult to understand/incomprehensible.

Reviewer 2 Report

Comments and Suggestions for Authors

--          define the abbreviation the first time they appear in the main text (for example, inflammatory bowel disease, Crohn’s disease, ulcerative colitis, etc.) and then always use the abbreviation

--          “Recent studies suggest that this condition may be linked to an imbalance in intestinal microorganisms or autoimmune factors”

IBD is not and autoimmune disease but an immunomediated disease

-  -        “gut flora”

Don’t use the old term flora but microbiota or microbiome

-   -       “These discoveries have caused a fundamental change in the approach to treating diseases, focusing on the goal of restoring the variety and equilibrium of microorganisms in the gut”

This is absolutely false: probiotics or FMT are not part of current therapy of IBD (see ECCO guidelines)

-    -      You have to redo your study because your systematic review missed several articles (for example: “Adalimumab Therapy Improves Intestinal Dysbiosis in Crohn's Disease. J Clin Med. 2019 Oct 9;8(10):1646. doi: 10.3390/jcm8101646. PMID: 31601034; PMCID: PMC6832711.”)

-     -     “However, since the composition of the microbiota is unlikely to affect the use of 5ASA therapy as the initial treatment”

Why? This is false. For example, “Mesalazine inhibits the growth of Mycobacterium avium subspecies paratuberculosis,   which has been reported to be intimately linked to the etiology of CD, in a dose-dependent manner in vitro. Another study has reported that mesalazine downregulates the gene expression that is associated with bacterial invasiveness and antibiotic resistance in Salmonella enterica serovar Typhimurium, which could promote the onset of IBD after its infection. Furthermore, mesalazine inhibits the growth of sulfate-reducing bacteria and suppresses sulfide production in IBD”

-      -    “The top left bar chart indicates significant variation in the number of participants across studies, with Oka and Sartor [43] including the largest sample size of 2495 participants”

Reference 43 is a review not an experimental article!! You have to include only original studies

-       -   “Clostridium difficile”

It was renamed into Clostridioides difficile several years ago

-        -  Always use italics for bacterial species (for example, “Faecalibacterium prausnitzii”)

-         - Use lower case for active ingredients (for example, vedolizumab)

Comments on the Quality of English Language

sufficient

Round 2

Reviewer 1 Report

Comments and Suggestions for Authors

Dear Authors,  

The comments raised during the first round of peer review have not been adequately addressed, particularly those related to the methodology and presentation of results, including tables and figures.  

Specific comments:  

Clearly organize the method steps and criteria for easy understanding.  

Data extraction is vague and needs clarification both in text and Table 1.  

Unclear how discrepancies were resolved?  

Be concise while maintaining the essential details. Avoid unnecessary repetition in results.  

Why statistical analysis included in results?  

I believe the results are bound to be flawed or questionable. Unclear how a total of 16 papers were selected for the qualitative analysis in figure 1.  

Clear justification why tables 2-3 are needed?  

All figures should be clearly organized.  

Avoid using the terms "we" and "our" throughout the manuscript (e.g., our understanding, we studies...etc.)

Comments on the Quality of English Language

Minor language editing (avoid using the terms we and our throughout the paper).

Reviewer 2 Report

Comments and Suggestions for Authors

Thank you for your corrections.

Comments on the Quality of English Language

Certifized.

Round 3

Reviewer 1 Report

Comments and Suggestions for Authors

Dear Authors,

I believe the statistical analysis should be moved to method after section 2.6. I will leave this to the Academic Editor, but I would not accept this.

Please used italics for genus name only. Firmicutes and Bacteroides are family names and should not be italicized.

Comments on the Quality of English Language

Minor language editing is required.
